# Bispecific Antibody-Based Immune-Cell Engagers and Their Emerging Therapeutic Targets in Cancer Immunotherapy

**DOI:** 10.3390/ijms23105686

**Published:** 2022-05-19

**Authors:** Ha Gyeong Shin, Ha Rim Yang, Aerin Yoon, Sukmook Lee

**Affiliations:** 1Department of Biopharmaceutical Chemistry, College of Science and Technology, Kookmin University, Seoul 02707, Korea; hgshin@kookmin.ac.kr (H.G.S.); 2825760@kookmin.ac.kr (H.R.Y.); 2R&D Division, GC Biopharma, Yongin 16924, Korea; 3Biopharmaceutical Chemistry Major, School of Applied Chemistry, Kookmin University, Seoul 02707, Korea; 4Antibody Research Institute, Kookmin University, Seoul 02707, Korea

**Keywords:** bispecific antibody, immune-cell engager, cancer, therapeutic target, T-cell, NK cell

## Abstract

Cancer is the second leading cause of death worldwide after cardiovascular diseases. Harnessing the power of immune cells is a promising strategy to improve the antitumor effect of cancer immunotherapy. Recent progress in recombinant DNA technology and antibody engineering has ushered in a new era of bispecific antibody (bsAb)-based immune-cell engagers (ICEs), including T- and natural-killer-cell engagers. Since the first approval of blinatumomab by the United States Food and Drug Administration (US FDA), various bsAb-based ICEs have been developed for the effective treatment of patients with cancer. Simultaneously, several potential therapeutic targets of bsAb-based ICEs have been identified in various cancers. Therefore, this review focused on not only highlighting the action mechanism, design and structure, and status of bsAb-based ICEs in clinical development and their approval by the US FDA for human malignancy treatment, but also on summarizing the currently known and emerging therapeutic targets in cancer. This review provides insights into practical considerations for developing next-generation ICEs.

## 1. Introduction

Cancer is one of the major leading causes of death worldwide. In 2020, nearly 19.3 million new cancer cases and 10.0 million cancer deaths were reported globally. More specifically, the most common cancers among new cases were breast cancer (11.7%), followed by lung (11.4%), colorectal (10.0%), prostate (7.3%), and stomach (5.6%) cancers [1]. The order of the mortality rate on the basis of cancer types was as follows: lung (18%), colorectal (9.4%), liver (8.3%), stomach (7.7%), and breast (6.9%) cancers [2]. By 2040, the number of new cancer cases is expected to be approximately 28.3 million, which is nearly >50% from that reported in 2020 [3]. Currently, various therapeutic regimens, such as surgical resection, chemotherapy, antibody therapy, radiotherapy, and combination therapy, have been used in clinical practice for the effective treatment of patients with cancers, depending on their health conditions and cancer status [4].

One of the most promising targeted therapies for cancer treatment is antibody therapy. It has a superior targeting ability for antigens that are expressed on cancer cells, which results in prominent antitumor activity and lower toxicity, compared with that of chemotherapeutic agents [5]. As of December 2021, 110 therapeutic antibodies, including monoclonal antibodies (mAbs), bispecific antibodies (bsAbs), and antibody–drug conjugates (ADCs), have been approved by the United States Food and Drug Administration (US FDA) and/or the European Medicines Agency (EMA). Among them, 46 antibodies are indicated for cancer treatment [6]. Generally, immunoglobulin (Ig) G-based mAb—the most widely used mAb form for antibody therapy—comprises two heavy and two light chains. The light chain has one variable (VL) and one constant (CL) domain, whereas the heavy chain has one variable (VH) and three constant (CH; CH1–CH3) domains [7,8]. Furthermore, the fragment antigen-binding (Fab) region of mAb plays a key role in cancer therapy, and specifically in modulating or blocking the signaling pathways that are involved in cancer development; on the other hand, the fragment crystallizable (Fc) region interacts with Fc receptors that are expressed on immune cells and participates in various effector functions, such as killing cancer cells via antigen-dependent T-cellular cytotoxicity (ADCC) [9,10].

BsAbs harness the specificities of two antibodies and combine them to simultaneously recognize two independent epitopes or antigens [11]. More specifically, these antibodies are designed and manufactured to contain two target-binding units in one antibody-based molecule, whereby each unit independently recognizes its unique epitope, through quadromas, chemical conjugation, or genetic recombination [12]. Compared with mAbs in cancer therapy, bsAbs have several potential benefits, such as the improvement in therapeutic efficacy, the enhancement of tumor-cell selectivity, and the reduction in tumor-cell resistance [13]. In 2009, catumaxomab—a rat–mouse hybrid bsAb for the CD3 epithelial cell adhesion molecule (EpCAM)—was first approved by the EMA for the treatment of malignant ascites in patients with EpCAM-positive cancer [14]. However, it was voluntarily withdrawn from the US market in 2013, and from the European Union (EU) market in 2017, because of commercial reasons [15]. Since the US FDA approval of blinatumomab—a CD19 × CD3 mouse bispecific T-cell engager (BiTE) antibody—for the treatment of acute lymphoblastic leukemia (ALL) in 2014, much attention has been paid to the development of bsAb-based immune-cell engagers (ICEs) that redirect immune effector cells against cancer cells and promote antitumor activities [16,17]. Furthermore, compared with adoptive immune-cell therapy, which requires an expensive and complicated manufacturing process, ICE has become a feasible therapeutic cancer-therapy approach. Currently, increasing numbers of bsAb-based ICEs are extensively evaluated in clinical trials worldwide [18,19].

Herein, we focus on highlighting the action mechanisms, structures, roles, and relevance of the currently known and potential therapeutic targets of bsAb-based ICEs in cancers. In addition, this review covers the current developmental status of bsAb-based ICEs approved by the US FDA and/or the EMA, or in clinical development for cancer therapy. This review provides insights into practical considerations for the development of the next-generation bsAb-based ICEs.

## 2. Action Mechanism of bsAb-Based ICEs in Cancer

BsAb-based ICEs play a key role in cancer immunotherapy, and specifically in recruiting and engaging immune effector cells that are proximal to tumor-associated antigens (TAA) that are expressed on cancer cells and that allow the formation of immune synapses and a specialized cell–cell junction between the immune and cancer cells [20,21]. Ultimately, these immune synapses promote the elimination of target cancer cells [22]. These bsAb-based ICEs are currently classified into T- and natural killer (NK)-cell engagers (Figure 1).

### 2.1. T-Cell Engagers

T-cell engagers are engineered bsAbs that redirect and activate T-cells to induce the robust elimination of poorly immunogenic tumors [23]. Most T-cell engagers comprise two linked antigen-binding variable fragments (Fvs) that specifically target TAAs and the CD3 unit of the T-cell-receptor (TCR) complex, which thereby engages T-cells to form an immune synapse on the surface of tumor cells [24,25]. Generally, the TCR complex on T-cells directly interacts with antigens that are presented on the major-histocompatibility-complex (MHC) molecules of target T-cells, and this interaction plays a pivotal role in T-cell activation and target T-cell killing [26]. However, most tumor cells are known to exhibit the loss or depletion of MHC expression, which hampers the antitumor effects of activated T-cells [27]. The unique feature of T-cell engagers is the redirection of T-cells against TAAs on tumor cells, as well as the direct activation of T-cells without TCR/MHC interaction [28]. Simultaneously, it links T-cells with tumor cells to form immune synapses, wherein the activated T-cells release perforins to form pores on the surface of cancer cells, and granzymes to proteolyze cellular proteins. Eventually, these immune responses lead to the cell death (also known as apoptosis) of tumors [22,29,30]. Moreover, T-cell activation by an anti-CD3 arm of a T-cell engager induces cytokine secretion and concomitant T-cell proliferation that sustain durable antitumor immune responses [31]. Currently, 47 bsAb-based T-cell engagers have been studied in clinical trials (Table 1).

### 2.2. NK-Cell Engagers

NK cells are effector lymphocytes of the innate immune system that play protective roles against both infectious pathogens and cancers [32]. NK cells are divided into several subpopulations on the basis of the relative expressions of the adhesion molecule CD56 and the activating receptor CD16 (FcγRIIIa) [33]. The CD56^dim^ CD16^bright^ NK cells represent at least 90% of all peripheral blood NK cells [34]. Currently, CD16 is the most important NK-cell target for the NK-cell engagers. CD16-targeting NK-cell engagers lead to both NK-cell activation and tumor-cell-specific cytotoxicity [35]. More specifically, NK-cell engagers recruit NK cells toward target tumor cells by binding a tumor-specific antigen with one arm of the engagers, and by bridging CD16 onto NK cells with the other [36]. Then, the NK-cell engagers trigger the death of target cancer cells by not only releasing cytotoxic granules that contain granzymes and perforin, but also by secreting chemokines and cytokines, such as those regulated on activation, normal T-cell expressed and secreted, and interferon-gamma [32]. AFM13 is a bsAb-based NK-cell engager that specifically targets CD16 and CD30; it is currently under clinical development for use in the treatment of hematological malignancy (Table 1).

## 3. Role of Known and Emerging Targets of ICEs 

### 3.1. Single-Pass ICE Targets in Solid Cancers

#### 3.1.1. Delta-Like Ligand 3

Delta-like ligand 3 (DLL3) is a 65-kDa type I transmembrane protein and a Notch receptor ligand. It plays an important role in the regulation of Notch signaling [37]. In small-cell lung cancer (SCLC), DLL3 has been reported as a key factor in the promotion of the tumor growth, migration, and invasion of SCLC cells. Several lines of evidence support this notion [37,38]. Upregulated DLL3 expression was verified to promote tumor growth in a mouse xenograft model that was implanted with DLL3-overexpressing SBC-5 human SCLC cells. Additionally, DLL3 knockdown reduces SCLC-cell migration and invasion, whereas its overexpression in the cells increases these activities [38]. This protein is highly upregulated and aberrantly expressed in SCLC and other neuroendocrine malignancies, but not in nonmalignant T-cells [37,39]. Currently, DLL3 is considered an attractive novel potential therapeutic target in neuroendocrine tumors (NETs), including SCLC. A preclinical study on robalpituzumab tesirin—an ADC that targets DLL3—showed a dose-dependent reduction in the tumor size with a complete response (CR) in B6129SF1/J mice that were implanted with DLL3-positive KP1 SCLC cells, which led to the absence of measurable tumors for >80 days after treatment [40].

#### 3.1.2. Epidermal Growth Factor Receptor

Epidermal growth factor receptor (EGFR) is a 170-kDa receptor tyrosine kinase that belongs to the ErbB family and that comprises two major functional domains—the extracellular and cytoplasmic domains—and a tyrosine kinase domain that is linked by a single transmembrane region [41,42]. EGF binding to the receptor induces the dimerization of the receptor; triggers the autophosphorylation of cytoplasmic tyrosine residues; and eventually participates in the regulation of cell proliferation, migration, and adhesion [41,43,44,45]. EGFR is overexpressed in various cancers, such as colorectal cancer (CRC), lung cancer, breast cancer, glioblastoma, and head and neck squamous cell carcinoma. EGFR overexpression in CRC has been closely associated with tumor progression and poor prognosis [46,47,48,49,50]. Currently, EGFR is one of the most well-known therapeutic targets in various cancers. Phase II clinical studies on cetuximab—a human/mouse chimeric mAb that targets EGFR in advanced CRC—have demonstrated that the use of cetuximab as monotherapy exerts anticancer effects with approximately 10% partial response (PR) and 33% stable disease (SD) [51].

#### 3.1.3. EpCAM

EpCAM is a 40-kDa type I transmembrane glycoprotein that plays a key role in the regulation of cell adhesion, proliferation, and differentiation [52,53,54]. EpCAM is overexpressed in various cancers, such as ovarian cancer, CRC, breast cancer, lung cancer, and pancreatic cancer [55,56,57,58]. Its protease-cleaved intracellular domain associates with β-catenin to form a nuclear protein complex that is translocated to the nucleus, activates the transcription of genes that are involved in cancer-cell proliferation, and results in tumorigenesis [52]. Several studies have suggested EpCAM as a potential target for antibody therapy against cancers. For instance, a phase II clinical study on adecatumumab (MT201)—a fully human mAb that targets EpCAM in metastatic breast cancer—reports that, of 112 patients treated with adecatumumab, 2 showed a PR and 10 had SD, according to the response evaluation criteria in solid tumors (RECIST) [59].

#### 3.1.4. Glycoprotein A33

Glycoprotein A33 (GPA33)—also known as cell surface A33 antigen—is a 43-kDa cell surface differentiation glycoprotein that belongs to the type I transmembrane protein family [60,61,62]. It is associated with cell–cell adhesion [60]. GPA33 is highly overexpressed in >95% of human CRCs but is not detected in any other tissues [62]. Several studies have indicated GPA33 as a potential target in immunotherapy against CRC. For instance, in vivo studies on KRN330—a human mAb that targets GPA33—have demonstrated its dose-dependent antitumor activities in mouse and rat xenograft models implanted with LS174T human CRC cells [63,64].

#### 3.1.5. Human EGFR 2

Human EGFR 2 (HER2) is a member of the ErbB family and is a 185-kDa single-pass transmembrane receptor. To the best of our knowledge, direct ligands for HER2 have not been identified yet. HER2 activation is achieved through homo- or heterodimerization with HER2 or other ErbB-family receptor members, including EGFR and HER3 [65,66,67]. It is overexpressed in various cancers, such as breast, gastric/gastroesophageal, and colon cancers [68,69,70]. HER2 is closely associated with cancer-cell proliferation and invasion, as well as with tumor growth [71,72]. Particularly in breast cancers, HER2 is overexpressed in 15–30% of the total patients with breast cancers [65]. Substantial evidence has shown that HER2 is an important predictive biomarker in HER2-targeted therapies, and a well-known therapeutic target in breast cancers. Trastuzumab is the first anti-HER2 humanized mAb that targets HER2 in breast cancer. In a phase III clinical study, patients with breast cancer treated with trastuzumab combined with chemotherapy had a longer survival (median survival, 25.1 vs. 20.3 months) and prolonged disease progression (median, 7.4 vs. 4.6 months) than those treated with chemotherapy alone [73].

#### 3.1.6. Mucin 16

Mucin 16 (MUC16)—also known as human carbohydrate antigen 125 (CA-125)—is a heavily glycosylated 300–500-kDa type I transmembrane protein [74,75,76]. It is a biomarker for ovarian cancer. MUC16 is overexpressed in ovarian cancer and contributes to ovarian-cancer progression and metastasis [76,77]. Increased MUC16 expression is associated with poor prognosis in patients with ovarian cancer [78]. Some studies have reported that MUC16 is a potential target for antibody therapy against ovarian cancers. Oregovomab is a mouse mAb that targets MUC16 in advanced ovarian cancer. In a phase II clinical study, 145 patients with stage III/IV ovarian cancer were randomized to receive oregovomab (*n* = 73) or placebo (*n* = 72). The time to recurrence was prolonged in the oregovomab group (24.0 months) compared with that in the placebo group (10.8 months) [79].

#### 3.1.7. Mucin 17

Mucin 17 (MUC17) is a 452-kDa type I membrane-associated mucin that is expressed on the apical surface of gastrointestinal epithelial cells [80,81,82]. As a key component of the mucosal layer, MUC17 has been suggested to play a crucial role in the restoration and protection of epithelial cells [80]. Recent studies have demonstrated that aberrant MUC17 overexpression is correlated with the malignant potential of gastric and pancreatic cancers [81,83]. Particularly in gastric-cancer tissues, MUC17 is overexpressed in approximately 50% of the gastric-cancer cases. Thus, MUC17 is a compelling target in gastric cancer because of its prevalent expression on tumor cells compared with its low, relatively restricted expression in normal tissues [84].

#### 3.1.8. Prostate-Specific Membrane Antigen

Prostate-specific membrane antigen (PSMA) is a 100-kDa type II membrane protein, is exclusively overexpressed in prostate cancer, and acts as a glutamate-preferring carboxypeptidase. Its expression is associated with tumor invasiveness [85,86]. PSMA is not only a well-known biomarker but is also a potential therapeutic target in prostate cancer. Its expression is 100–1000-fold higher in prostate-cancer tissue than in normal tissue, and it is present on the cell surface without being released into the circulation [87,88,89,90]. Furthermore, J591 was recently developed as the first humanized mAb that targets the extracellular domain of PSMA in prostate cancer. A phase I/II study was conducted to evaluate the safety and efficacy of J591 in patients with metastatic castration-resistant prostate cancer (mCRPC). Of the 23 patients with measurable disease, 14 (60.8%) had SD and 6 (26.1%) had progressive disease, according to the RECIST [91].

### 3.2. Multi-Pass Transmembrane Proteins as ICE Targets in Solid Cancers

#### 3.2.1. Claudin-18 Isoform 2

Claudin-18 isoform 2 (CLDN18.2) is a 23-kDa tetra-transmembrane protein. It plays an important role in the regulation of tight junction formation and cell adhesion [92,93,94]. It is known as a tumor-specific marker in gastric or gastroesophageal junction (GEJ) cancers because it is overexpressed exclusively in primary gastric malignancies, but not in any healthy tissues, except stomach mucosa [90,92,95]. Some studies have suggested that CLDN18.2 is a target for antibody therapy against cancer. Claudiximab (IMAB362) is a chimeric mAb that targets CLDN18.2 in gastric cancer. In a phase II study, patients with advanced/recurrent gastric and GEJ cancers who were treated with claudiximab combined with chemotherapy exhibited a significantly improved progression-free survival (PFS) (median, 7.9 vs. 4.8 months) and prolonged overall survival (OS) (median, 13.3 vs. 8.4 months) compared with those treated with chemotherapy alone [96].

#### 3.2.2. Six-Transmembrane Epithelial Antigen of Prostate 1

Six-transmembrane epithelial antigen of prostate 1 (STEAP1) is a 39-kDa integral membrane protein that comprises six transmembrane helices [97,98]. In normal cells, STEAP1 plays a key role in the regulation of cell migration and proliferation, despite its low expression or absence in normal tissues [97,99]. It is highly overexpressed in various cancers, and particularly in prostate cancer, wherein it is involved in the regulation of various functions, such as cancer-cell invasion and proliferation, as well as tumorigenesis [98,99]. The knockdown of STEAP1 has been shown to inhibit T-cell growth in androgen-dependent prostate cancer [100]. Moreover, high STEAP1 expression is closely associated with poor outcomes in patients with prostate cancer [101]. These properties make STEAP1 a potential target for antibody therapy. DSTP3086S—an ADC that targets STEAP1—exhibited antitumor activity in a phase I clinical trial in patients with mCRPC. Of the 46 patients, 2 (4%) showed a PR and 24 (52%) had SD, according to the RECIST [102].

#### 3.2.3. Somatostatin Receptor 2

Somatostatin receptor 2 (SSTR2) is a 41-kDa G protein-coupled receptor (GPCR), which is also known as a seven-transmembrane receptor. It is highly overexpressed in most NETs [103,104,105]. Among the NETs, and particularly in SCLC, the high expression of SSTR2 is closely associated with poor prognosis. Furthermore, the loss of SSTR2 reduced tumor growth in a mouse xenograft model implanted with H1048 human SCLC cells [104]. Several studies have suggested that SSTR2 is a therapeutic target in NETs. For instance, the antitumor efficacy of ADC that targets SSTR2 was evaluated in a mouse xenograft model implanted with BON-1 human NET cells, in which it reduced tumor growth [106].

### 3.3. Glycosylphosphatidylinositol-Anchored Proteins as ICE Targets in Solid Cancers

#### 3.3.1. Carcinoembryonic Antigen

Carcinoembryonic antigen (CEA) is a 180–200-kDa member of the immunoglobulin supergene family. It plays a key role in the regulation of various cellular functions, such as cell interaction, cell adhesion, and immune response [107,108,109]. CEA is one of the most widely used tumor-marker proteins for various cancers, such as colorectal, gastric, and liver cancers [110,111,112]. It is highly overexpressed in 90% of the total CRC cases, and it is closely associated with poor prognosis in patients with CRC [113]. In CRCs, CEA is involved in cancer progression and metastasis, as well as drug resistance [110,114,115,116,117]. Furthermore, CEA appears to be a potential target for antibody therapy against CRC. A preclinical study on IMMU-130—an ADC that targets CEA—revealed that the ADC efficiently reduced tumor growth in a mouse xenograft model implanted with LS174T human CRC cells [118,119].

#### 3.3.2. Glypican 3

Glypican 3 (GPC3) is a 60-kDa glycosylphosphatidylinositol (GPI)-anchored membrane-bound heparin sulfate proteoglycan. It plays an important role in normal cell growth [120,121]. GPC3 is overexpressed in various cancers, such as hepatocellular carcinoma (HCC), lung squamous cell carcinoma, and ovarian clear cell carcinoma [122,123,124]. In particular, it is highly overexpressed in 70–81% of HCCs; its overexpression correlates with the poor prognosis of patients with HCC [121]. Several studies have suggested that GPC3 is a target for antibody therapy against HCCs. For instance, a preclinical study on GC33—a mAb that targets GPC3—demonstrated its prominent antitumor activity in a mouse xenograft mouse model implanted with SK-HEP-1 human HCCs. The administration of 1 mg/kg of GC33 significantly inhibited tumor growth, and that of 5 mg/kg resulted in tumor remission [125].

### 3.4. Sphingolipid as ICE Targets in Solid Cancers

#### GD2

GD2 is a 1.6-kDa glycosylated lipid molecule that belongs to the class of glycosphingolipids [126,127,128,129]. It plays a key role in the attachment of tumor cells to extracellular matrix proteins. It is overexpressed in various cancers, such as neuroblastoma, melanoma, and SCLC, but not in normal tissues [127,130,131,132]. Particularly in SCLC and neuroblastoma, GD2 overexpression is involved in cell proliferation [130]. GD2 has been suggested as a target for antibody therapy against cancer. A phase II clinical study on 3F8—a mouse mAb that targets GD2 in patients with neuroblastoma—revealed that, of 16 patients, 1 showed a CR, and 1 showed a mixed response [133].

### 3.5. Single Transmembrane Proteins as ICE Targets in Hematological Cancers

#### 3.5.1. B-Cell Maturation Antigen

B-cell maturation antigen (BCMA)—a member of tumor necrosis factor receptor superfamily member 17 (TNFRSF17)—is a 20-kDa type III transmembrane protein [134]. BCMA binds to its ligands, such as proliferation-inducing ligand and B-cell-activating factor, which thus promotes the survival of B-cells [135,136]. It is overexpressed in malignant plasma cells, including multiple myeloma (MM) cells, and it plays a crucial role in the growth of MM [137,138]. A preclinical study that was conducted that used a mouse xenograft model implanted with RPMI 8226 human MM cells has shown that BCMA overexpression promotes tumor growth [137]. Furthermore, high BCMA expression is associated with poor prognosis in patients with MM [139]. Substantial evidence has shown that BCMA is a target for antibody therapy against MM. Preclinical studies on belantamab mafodotin—a US FDA-approved ADC that targets BCMA—showed that it efficiently inhibited tumor growth and prolonged survival in a mouse xenograft model implanted with H929 human MM cells [140,141].

#### 3.5.2. CD19

CD19 is a 95-kDa type I transmembrane protein [142]. It is a coreceptor of B-cell antigen receptor (BCR), and it plays a role in regulating B-cell growth [143,144]. It is overexpressed in most B-cell malignancies, such as ALL, non-Hodgkin lymphoma (NHL), and chronic lymphocytic leukemia (CLL). The overexpression of CD19 promotes the proliferation and survival of these B-cell malignancies [145,146]. Previous studies have suggested that CD19 is an attractive target for antibody therapy against B-cell malignancies. In a phase IIa clinical study on XmAb5574 (MOR00208)—a humanized mAb that targets CD19—of the total patients with relapsed and/or refractory (R/R) NHL who received XmAb5574 monotherapy, 8% showed a CR [147].

#### 3.5.3. CD22

CD22 is a 140-kDa type I transmembrane protein and an inhibitory coreceptor of the BCR that regulates the overstimulation of B-cells [148,149]. CD22 is overexpressed in various B-cell lymphomas, such as CLL, ALL, and NHL, but it is expressed at low levels on immature B-cells and plasma cells [150,151]. Owing to the restricted expression on the B-cell and the inhibitory function of CD22, CD22 has been indicated as a therapeutic target in B-cell lymphoma [152]. Epratuzumab—a humanized mAb that targets CD22—has been reported to be a CD22 agonistic antibody that leads to B-cell inhibition [153]. In a phase II clinical trial, patients with R/R indolent or aggressive NHL were enrolled to receive epratuzumab combined with rituximab, which is a US FDA-approved anti-CD20 mAb. Of the 16 patients with indolent NHL, 9 showed a CR and an unconfirmed CR, and 1 showed a PR. Furthermore, of the six patients with aggressive NHL, three showed a CR, and one showed a PR [154,155].

#### 3.5.4. CD30

CD30 is a 120-kDa type I transmembrane protein that belongs to the tumor necrosis factor receptor family [156]. It plays a key role in lymphocyte activation and proliferation through the nuclear factor-kappa B (NF-κB) and mitogen-activated protein kinase pathways that have antiapoptotic and prosurvival benefits [157,158]. It is overexpressed in hematopoietic malignancies, including Hodgkin lymphoma (HL) and NHL, and is associated with the survival of these cells [159,160]. Several studies have suggested that CD30 is a target for antibody therapy against hematologic malignancies. For instance, in vivo studies on XmAb2513—a humanized mAb that targets CD30—showed a significant reduction in the tumor growth, and enhanced survival was observed in a mouse xenograft model implanted with CD30-expressing L540 human HL cells [161].

#### 3.5.5. CD33

CD33—also known as the sialic acid-binding Ig-like lectin 3 (Siglec-3)—is a 67-kDa type I transmembrane protein [162]. It plays a crucial role in the modulation of immune-cell functions, such as phagocytosis, cytokine release, and apoptosis [163,164]. It is overexpressed in acute myeloid lymphoma (AML), and its overexpression is observed in >80% of patients with AML [165]. This increased CD33 expression is correlated with the poor prognosis of patients with AML. In patients with AML who were treated with chemotherapy, the OS rate has been reported to be 42.9% in patients with high CD33 expression, compared with 67.5% in those with low CD33 expression [166]. CD33 is a target for antibody therapy against AML. Gemtuzumab ozogamicin (Mylotarg)—a US FDA-approved ADC that targets CD33—showed promising clinical efficacy in patients with AML [167]. In a phase II clinical study, patients with AML in the first recurrence received gemtuzumab ozogamicin monotherapy; of the 277 patients, 35 showed a CR, and 36 showed a CR with incomplete platelet recovery [168].

#### 3.5.6. CD38

CD38 is a 45-kDa type II transmembrane protein [169]. It is overexpressed in MM cells but shows a low expression in normal lymphoid and myeloid cells [170]. It participates in MM cell survival and proliferation [171]. Previous studies have elucidated that tumor growth decreased in a mouse xenograft model implanted with CD38-knockout RPMI 8226 human MM cells, compared with those nontargeting cells [172]. CD38 is a potential target for antibody therapy against MM. Daratumumab (Darzalex) is the first US FDA-approved human mAb that targets CD38 in the treatment of patients with R/R MM [173]. In a phase III clinical study, patients with R/R MM received chemotherapy (control group) or chemotherapy combined with daratumumab (daratumumab group); the CR rate was significantly higher in the daratumumab group (19.2%) than in the control group (9.0%) [174].

#### 3.5.7. CD123

CD123—the alpha chain of the interleukin-3 (IL-3) receptor—is a 75-kDa type I transmembrane protein [175]. It is overexpressed in leukemic stem cells but shows low or no expression in normal hematopoietic stem cells [176]. CD123 binds to IL-3, which is a hematopoietic growth factor, which leads to the survival and proliferation of various hematologic cancers, such as AML, ALL, and HL [175,177,178,179]. Particularly in AML, increased CD123 expression is associated with a poor prognosis of patients with AML [180]. Previous studies have suggested that CD123 is a target for antibody therapy against AML. In vivo studies on IMGN632—an ADC that targets CD123—have revealed its antitumor activities in a mouse xenograft model implanted with MOLM-13 human AML cells; the mice received IMGN632 or control ADC, and IMGN632 increased the survival of mice compared with the vehicle treatment [181].

#### 3.5.8. C-Type Lectin Domain Family 12 Member A

C-type lectin domain family 12 member A (CLEC12A)—a myeloid inhibitory receptor—is a 31-kDa type II transmembrane protein. It plays a crucial role in the negative regulation of inflammation [182,183]. CLEC12A is specifically expressed in AML and is observed in approximately 90% of patients with AML, but not in normal hematopoietic stem and progenitor cells [184,185]. It is closely associated with the poor prognosis of patients with AML. Previous studies have shown that CLEC12A-positive AML cells are more resistant to chemotherapy than CLEC12A-negative AML cells [186]. Furthermore, the administration of anti-CLEC12A chimeric mAb showed a significant tumor-growth delay of up to 38% in a mouse xenograft model implanted with HL-60 human AML cells [187]. These observations suggest that CLEC12A is a target for antibody therapy against AML.

#### 3.5.9. FMS-Like Tyrosine Kinase 3 

FMS-like tyrosine kinase 3 (FLT3)—a receptor tyrosine kinase—is a140–160-kDa type I transmembrane protein [188]. FLT3 promotes the proliferation and differentiation of hematopoietic cells by binding to its ligand [189,190]. FLT3 is overexpressed in AML, and its mutations have been detected in approximately 30% of patients with AML [191,192]. The mutation of *FLT3* causes ligand-independent FLT3 signaling and leads to a poor prognosis of patients with AML [193,194]. FLT3 is a potential target for antibody therapy against AML. LY3012218 (IMC-EB10) is a human mAb that targets FLT3, which prevents FLT3 signaling [195]. In preclinical studies, LY3012218 has shown efficacy in a mouse xenograft model implanted with MOLM-14 human AML cells; LY3012218 exerts its effects by reducing the engraftment of leukemic cells and extending survival [196].

### 3.6. Multi-Pass Transmembrane Proteins as ICE Targets in Hematological Cancers

#### 3.6.1. CD20

CD20 is a 33–37-kDa tetra-transmembrane protein [197]. It is involved in B-cell differentiation and it is overexpressed in most B-cell malignancies, such as follicular lymphoma (FL), but not in hematopoietic stem cells or plasma cells [198,199,200]. Several studies have shown that CD20 is a potential target for antibody therapy against FL. Rituximab (Rituxan) is a chimeric mAb that has been approved by the US FDA against CD20 [155]. In a phase III clinical study, patients with R/R FL received lenalidomide plus rituximab or placebo plus rituximab; the median PFS increased in the lenalidomide-plus-rituximab group (39.4 months), compared with that in the placebo-plus-rituximab group (14.1 months) [201].

#### 3.6.2. GPCR Class C Group 5 Member D

GPCR class C group 5 member D (GPRC5D) is a 39-kDa seven-transmembrane protein [202]. It is an orphan receptor that is normally expressed only in the hair follicle. GPRC5D is overexpressed in MM and is unlikely to be shed from the membrane, which prevents the decrease in the efficacy of GPRC5D-targeted therapy [203,204,205]. The role of GPRC5D in cancers is yet to be defined; nonetheless, selective GPRC5D expression may be valuable as a target for antibody therapy against MM. Figure 2 summarizes the known and emerging targets for ICE therapy against cancers.

## 4. Design and Structure of bsAb-Based ICEs

BsAb-based ICEs are designed to contain two different antigen-binding sites that comprise determinants from the VH and VL chains of different antibodies that are specific to each target [206]. Thus far, various efforts have been made to increase their homogeneity, yield, and functional properties to generate desired bsAb-based ICEs [11]. On the basis of the bsAb-based ICE structures, they are divided into two categories: Fv-based ICEs and immunoglobulin G (IgG)-based ICEs (Figure 3). Fv-based ICEs are easy to produce and show lower immunogenicity, whereas IgG-based ICEs have higher solubility, stability, affinity, and extended half-life in serum [207].

### 4.1. Fv-Based ICEs

#### 4.1.1. BiTE

BiTE comprises two single-chain Fvs (scFvs) combined with a flexible linker [29,208]. Blinatumomab (Blincyto) was the first US FDA-approved BiTE against CD19 and CD3 for the treatment of patients with R/R B-cell ALL [209]. It has a molecular weight of approximately 55 kDa and it is constructed by linking anti-CD19 scFv in a VL–VH orientation to anti-CD3 scFv in a VH–VL orientation through a short polypeptide (G4S) linker [29,208]. Furthermore, each scFv has two flexible long (G4S)3 linkers between the VH and VL to maintain the proper conformation of scFvs [210].

#### 4.1.2. Dual-Affinity Retargeting Protein

Dual-affinity retargeting protein (DART) comprises two engineered heterogenous scFvs, which exchanged their VH regions, and it has a molecular weight of approximately 50 kDa [211]. Precisely, scFv1 comprises a VH from antibody A and a VL from antibody B, and scFv2 comprises a VH from antibody B and a VL from antibody A in the VL(B)–VH(A) and VL(A)–VH(B) orders, respectively [212]. This combination allows DART to mimic natural interactions within IgG molecules. The scFv1 of flotetuzumab—a CD3 × CD123 DART—comprises VL(CD123)—linker—VH(CD3), whereas scFv2 comprises VL(CD3)–linker–VH(CD123) [213]. The linker between VH and VL of the DART platform is as short as approximately five amino acids to prevent their association from forming an undesired scFv. Moreover, the C-terminal disulfide bridge between two VHs contributes to holding the molecule together in the correct orientation [214]. CD19 × CD3 DART molecules have been reported to be more stable and potent than CD19 × CD3 BiTE molecules in targeting and killing B-cell lymphoma cells [210,212,214].

#### 4.1.3. Tandem Diabody

Tandem diabody (TandAb) is a novel tetravalent bsAb-based ICE with four binding sites: two for tumor antigens and the other two for immune cells [215]. It has a molecular weight of approximately 105 kDa. The configuration of TandAb is as follows: VH(A)–linker 1–VL(B)–linker 2–VH(B)–linker 3–VL(A) [216]. In this configuration, both linkers 1 and 3 comprise six amino acids (GGSGGS); however, linker 2 has one of the following three peptide sequences: GGSG, GGSGG, or GGSGGS [213]. AFM13—a CD16- and CD30-specific NK-cell engager—is currently being evaluated in a phase II clinical study for use in the treatment of patients with CD30-positive T-cell lymphoma [217].

### 4.2. IgG-Based ICEs: Symmetric Format

IgG-based ICEs are classified into symmetric and asymmetric architecture. Symmetric formats are mainly tetravalent (2 + 2) and are constructed by Fv or Fab fused with an IgG molecule [11].

#### 4.2.1. Fabs-In-Tandem Ig

Fabs-In-Tandem Ig (FIT-Ig) is a tetravalent ICE, where Fab(A) is structurally fused in tandem with the N terminus of Fab(B), without any mutations or the use of peptide linkers [218]. It has a molecular weight of approximately 240 kDa. More specifically, it comprises one long chain and two short chains: the long chain where the light-chain (VL(A)–CL) domains are directly fused in tandem with the N terminus of a heavy chain (VH(B)–CH1–CH2–CH3), and two short chains (VH(A)–CH1 and VL(B)–CL) [218,219]. The resulting FIT-Ig may have activities that are similar to those of both parental mAbs [218]. EMB-06a—a CD3 × BCMA FIT-Ig—combines intact Fab fragments from two parental antibodies, and it exhibits favorable drug-like properties and manufacturing advantages that are similar to those of each parental mAb [218,220].

#### 4.2.2. IgG–[L]–scFv2

IgG–[L]–scFv2 is a tetravalent ICE that is constructed by fusing the scFv with the C terminus of each IgG light chain [221]. This antibody is constructed by linking the human CD3-specific scFv on T-cells to the C terminus of the light chain of each antitumor IgG via a polypeptide linker [222,223]. Nivatrotamab is a 200-kDa (nearly) type of IgG–[L]–scFv2 antibody [224]. The heavy chain of nivatrotamab is identical to that of an anti-GD2 IgG, whereas its light chain is constructed by linking an anti-GD2 IgG light chain to a (G4S)3 linker, followed by the CD3-specific scFv [222,223,224]. Previous studies have indicated IgG–[L]–scFv2 as a promising platform for robust antitumor activity [221].

#### 4.2.3. IgG–[H]–scFv2

IgG–[H]–scFv2 is a tetravalent ICE that is constructed by fusing the scFv with the C terminus of each IgG heavy chain. In this format, CD3-specific scFvs are covalently attached to the C terminus of each TAA-specific IgG heavy chain [210]. CC-1 (CD3 × PSMA) and CLN-049 (CD3 × FLT3) are the major ICEs of IgG–[H]–scFv2, with a molecular weight of approximately 200 kDa [225]. To construct the CC-1 ICE, anti-CD3 scFv is linked to an anti-PSMA IgG by a flexible (G4S)3 linker [226]. CC-1 ICE has been reported as a highly potent ICE with significant productivity and low aggregation [227].

#### 4.2.4. scFv2–Fc–scFv2

scFv2–Fc–scFv2, including Aptevo’s ADAPTIR antibodies, is a type of tetravalent ICE that comprises two scFv pairs that are joined via the Fc region [226]. Antitumor scFvs are attached to the N terminus of the hinged domain, whereas anti-CD3 scFvs are fused at the C terminus of the Fc region with an unknown linker. The length and composition of the ADAPTIR linker vary to modulate the binding and activity [228]. APVO436 (CD3 × CD123), which is generated by using the ADAPTIR platform as a T-cell engager, is currently being evaluated in clinical studies [229].

### 4.3. IgG-Based ICEs: Asymmetric Format

The major challenge in the generation of IgG-based ICEs with asymmetric formats is ensuring the correct association between the heavy and light chains [11]. Thus far, various conventional and state-of-the-art technologies, including mouse–rat hybrid, knob-into-hole (KiH), charge pair, controlled Fab-arm exchange (cFAE), common light chain, and CrossMab technology, have been used to develop desirable IgG-based ICEs to avoid random associations between the heavy and light chains [230,231,232].

#### 4.3.1. Mouse–Rat Hybrid IgG

Mouse–rat hybrid IgG is a bispecific chimeric antibody that is generated by using the quadroma technology that is based on the fusion of two distinct hybridomas [233]. This antibody is also called the trifunctional antibody or Triomab, owing to the retained effector function of the mouse/rat Fc part [234]. It comprises two different full-size IgG-like half antibodies, each with one light and one heavy chain, which originate from the parental-mouse-IgG2a and rat-IgG2b isotypes [235]. The use of the isotype combination enables the high yield of correctly paired bsAbs because of species-restricted heavy–light chain pairing [236]. By using this technology, catumaxomab—an EpCAM- and CD3-specific ICE—was developed and approved as the first bispecific ICE in 2009 by the EMA for the treatment of patients with malignant ascites [235]. However, catumaxomab was reported to have several issues that are related to the highly immunogenic nature of the mouse–rat hybrid antibody and large-scale commercial manufacturing. Catumaxomab was voluntarily withdrawn from the US and EU markets in 2013 and 2017, respectively [237].

#### 4.3.2. KiH-Based IgG

The KiH technology invented by Genentech is the most widely used bispecific platform for the generation of asymmetric bsAbs [238]. The concept relies on interface modifications between the two CH3 domains that are crucial for reciprocal interactions. A bulky residue is introduced as a key into the CH3 domain of one antibody heavy chain, and a hole that accommodates this bulky residue acts as a lock in the other heavy chain. Precisely, a knob is designed by replacing T366 with a bulky W residue on one heavy chain, and the three amino acid residues on the partner heavy chain are changed into T366S, L368A, and Y407V for the hole formation [238,239]. This technology produces >90% of the desired asymmetric bsAbs under coexpression conditions [240]. More recently, the yield of the heterodimeric bsAb has been increased to >97% by the introduction of two additional mutations: S354C in the knob chain, and Y349C in the hole chain [241]. Furthermore, these KiH mutations are known to not significantly affect the antibody properties, such as the immunogenicity, thermal stability, FcɤR binding, Fc effector function, and pharmacokinetic behavior [235].

#### 4.3.3. Charge-Pair-Based IgG

The charge-pair-based technology is a novel platform to generate electrostatically matched Fc domains by altering the charge polarity between the CH3 interfaces of an IgG antibody [242]. Previous studies have reported that the point mutations of three specific charged residue pairs in the heavy chains (E356K-K439E, E357K-K370E, and D399K-K409D) resulted in favorable attractive interactions [235]. This method uses the asymmetric re-engineering technology Ig (ART-Ig) platform [242]. ERY974, which is a CD3- and GPC3-specific ICE based on the ART-Ig platform, comprises E365K in one anti-CD3 heavy chain, and K439E in the other anti-GPC3 heavy chain [243]. However, the charge-pair approach may not result in the higher yield and purity of the respective heterodimeric antibodies than the KiH approach [244].

#### 4.3.4. cFAE-Based IgG

cFAE technology is inspired by a natural immune phenomenon that is known as Fab-arm exchange, in which a half molecule of an IgG4 antibody is reassembled with that of another IgG4 antibody in vivo [232,245]. It is used to generate IgG-like bsAbs. One pair of these matching CH3 mutations (F405L in one antibody and K409R in the other) has been reported through extensive site-directed mutagenesis in IgG1 [246]. The cFAE platform does not need additional technology to correct the light-chain assembly because light-chain mispairing does not occur in this case [241]. Epcoritamab—a CD3 × CD20 ICE—is generated by individually introducing an F405L mutation into an anti-CD3 heavy chain, and a K409R mutation into a CD20-specific heavy chain [247]. This technology includes the DuoBody platform that was developed by Genmab. Recently, amivantamab—a bispecific molecule that targets EGFR and mesenchymal–epithelial transition factor (c-MET)—which was produced by Johnson & Johnson by using the aforementioned platform, was approved by the US FDA for the treatment of patients with locally advanced or metastatic non-SCLC [248].

#### 4.3.5. Common Light-Chain-Based IgG

Common light-chain technology is developed to overcome the mispairing of light chains and is used in combination with an Fc-modified technology, such as KiH and charge-pair-based technologies [249]. This technology is based on the findings that antibodies against various antigens often share the same VL domain, in which antibodies were identified from phage display libraries with a very limited size of the light-chain repertoire [210]. bsAbs that are designed by using the common light-chain technology have identical light chains that can be paired with two different heavy chains. This antibody can prevent the mispairing between light and heavy chains [238]. By using this technology, REGN1979—a CD20- and CD3-specific ICE—was developed and is currently being evaluated in a phase II clinical study for use in the treatment of patients with B-cell NHL [250]. The common light-chain technology is based on simplifying antibody engineering, but this approach may cause less flexibility for antibody engineering [210].

#### 4.3.6. CrossMab-Based IgG

CrossMab technology is developed to inhibit the mispairing of light chains [251]. However, it takes a different approach from that of the common light-chain technology. This technology is based on the crossover of the constant region (CL and CH1 domains) of one Fab arm, whereas the other Fab arm undergoes no change [241]. This approach employs the KiH technology for correct heavy-chain pairing, as well as domain swap to enable orthogonal light–heavy chain pairing [207]. Glofitamab—a CD20- and CD3-specific ICE—was developed by using this technology and is currently being evaluated in a phase III clinical study for use in the treatment of patients with diffuse large B-cell lymphoma (DLBCL) [252].

#### 4.3.7. Fab–scFv–Fc

The Fab–scFv–Fc format is another strategy to correct the pairing of the light chain. The issue of light-chain mispairing is resolved by exchanging one Fab arm with scFv [253]. Fab–scFv–Fc comprises one Fab arm and one scFv that is fused with an Fc domain [254]. This antibody is used by several companies and it has various names, such as Xmab, Ybody, and BEAT [254,255]. Tidutamab (XmAb18087)—a CD3- and SSTR2-specific ICE—is currently being evaluated in a phase I/II clinical study for use in the treatment of patients with SCLC [256]. Additionally, six different ICEs that have the Fab–scFv–Fc format are currently being evaluated in clinical studies [257,258].

#### 4.3.8. Fc-Fused Fvs

The small size of Fv-based ICEs contributes to high renal clearance, which results in a half-life shorter than that noted in the case of IgG-based ICEs. BiTE and DART ICEs have recently been engineered to improve their pharmacokinetics (PK) [210]. Several BiTEs are linked to the Fc domain to generate half-life-extended BiTE-Fc molecules (HLE-BiTE), which are compatible with once-weekly dosing for treatments. DART is fused with the Fc region, which generates a DART–Fc complex. MGD007 (CD3 × GPA33) was constructed as a DART–Fc complex that results in a significant extension of the half-life in serum [259].

## 5. Current Development Status of bsAb-Based ICEs 

### 5.1. ICEs Targeting Tumor Antigens in Solid Cancers

#### 5.1.1. CEA-Specific ICE

RO6958688 is a CD3- and CEA-specific bsAb-based T-cell engager that is used for the treatment of patients with advanced CEA-positive solid tumors. In phase I clinical studies (NCT02324257 and NCT02650713), RO6958688 has been administered as monotherapy or in combination with atezolizumab—a US FDA-approved anti-PD-ligand 1 mAb—in patients with advanced CEA-positive solid tumors. Two patients showed a PR after monotherapy, and two after its use in combination with atezolizumab. Antitumor activity was observed with RO6958688 monotherapy, which was enhanced when this engager was used in combination with atezolizumab, with a manageable safety profile [260].

#### 5.1.2. CLDN18.2-Specific ICE

AMG 910 is a bsAb-based T-cell engager against CD3 and CLDN18.2 that is used for the treatment of patients with gastric or GEJ adenocarcinoma [261]. A phase I clinical study (NCT04260191) on AMG 910 is currently being conducted to evaluate its dose-limiting toxicity, treatment-related adverse event, PK, response duration, time to progression, and recommended phase 2 dose (RP2D) [261,262].

#### 5.1.3. DLL3-Specific ICE

Tarlatamab (AMG 757) is a CD3- and DLL3-specific bsAb-based T-cell engager that is used for the treatment of patients with DLL3-expressing SCLC. In a phase I clinical study (NCT03319940) on tarlatamab, a confirmed PR was reported across all dose levels in 8 out of 60 patients (13%), and an unconfirmed PR in 5 out of 8 patients (63%), at the highest dose level [263]. A phase II clinical study (NCT05060016) is currently ongoing to evaluate the safety and efficacy of tarlatamab [264].

#### 5.1.4. EGFR-Specific ICE

REGN7075 is a CD28- and EGFR-specific bsAb-based T-cell engager that is used for the treatment of patients with advanced solid cancers. A phase I/II clinical trial (NCT04626635) is currently being conducted to assess the safety and tolerability of REGN7075 in combination with cemiplimab—a US FDA-approved antihuman programmed cell death receptor-1 (PD-1) mAb—in patients with advanced solid tumors [265].

#### 5.1.5. EpCAM-Specific ICE

Catumaxomab (Removab) is the first bsAb-based T-cell engager that was approved by the EMA against CD3 and EpCAM for use in the treatment of malignant ascites in patients with EpCAM-positive cancer [14]. In a phase II/III clinical study (NCT00836654), the puncture-free survival was significantly longer in the catumaxomab group (median: 46 days) than in the control group (median: 11 days) [266]. M701 is another type of CD3- and EpCAM-specific bsAb-based T-cell engager that is used for the treatment of malignant ascites that are caused by advanced solid tumors. The interim results of a phase I clinical study (NCT04501744) on M701 revealed promising efficacy results: of 16 patients treated with M701, 3 showed a CR, 7 showed a PR, and 6 had SD [267].

#### 5.1.6. GD2-Specific ICE

Nivatrotamab is a CD3- and GD2-specific bsAb-based T-cell engager that is used for the treatment of patients with metastatic SCLC [268]. A phase I/II clinical study (NCT04750239) is currently being conducted to evaluate the safety and tolerability of nivatrotamab in patients with metastatic SCLC [269].

#### 5.1.7. GPA33-Specific ICE

MGD007 is a bsAb-based T-cell engager against CD3 and GPA33 that is based on the DART platform and that is used for the treatment of patients with metastatic CRCs [270]. The safety, tolerability, and efficacy of MGD007 in combination with retifanlimab (previously known as MGA012)—an investigational anti-PD-1 humanized mAb—were assessed in a phase I/II clinical study (NCT03531632) [271].

#### 5.1.8. GPC3-Specific ICE

ERY974 is a CD3- and GPC3-specific bsAb-based T-cell engager that is used for the treatment of patients with advanced or metastatic HCCs [272]. A phase I clinical study (NCT05022927) is currently being conducted to evaluate the safety, tolerability, PK, and antitumor activity of ERY974 in combination with atezolizumab and bevacizumab—a US FDA-approved anti-vascular endothelial growth factor mAb—in patients with locally advanced or metastatic HCC [273].

#### 5.1.9. HER2-Specific ICE

M802 is a bsAb-based T-cell engager against CD3 and HER2 that is used for the treatment of patients with HER2-positive advanced solid cancers [255,274]. M802 has been approved for phase I clinical study (NCT04501770) to evaluate its safety and tolerability in patients with HER2-positive advanced solid cancers [274].

#### 5.1.10. MUC16-Specific ICE

Ubamatamab (REGN4018) is a bsAb-based T-cell engager against CD3 and MUC16 that is used for the treatment of patients with advanced ovarian cancer. A phase I/II clinical study (NCT03564340) is currently being conducted to assess the PK, safety, tolerability, and preliminary antitumor activity of ubamatamab monotherapy and its combined use with cemiplimab, which is a US FDA-approved anti-PD-1 mAb [275].

#### 5.1.11. MUC17-Specific ICE

AMG 199 is an HLE-BiTE molecule against CD3 and MUC17. It induces the activation and proliferation of CD3^+^ T-cells and activates T-cell-mediated tumor-cell lysis in MUC17-positive gastric cancer [81]. A phase I clinical study (NCT04117958) on AMG 199 is currently being conducted to evaluate the safety, tolerability, PK, and antitumor activity of AMG 199 in MUC17-positive solid tumors, and to determine the maximum tolerated dose and/or RP2D [276].

#### 5.1.12. PSMA-Specific ICE

Acapatamab (AMG 160) is a CD3- and PSMA-specific bsAb-based T-cell engager that is used for the treatment of patients with mCRPC. Acapatamab induces the infiltration, activation, and expansion of T-cells, as well as the killing of tumor cells [277]. A phase I clinical study (NCT03792841) is currently being conducted to evaluate the safety and tolerability of acapatamab monotherapy or its combined use with pembrolizumab (Keytruda)—an anti-PD-1 mAb—in patients with mCRPC [278]. Additionally, other anti-PSMA bsAb-based ICEs, such as CC-1, JNJ-63898081, TNB-585, REGN5678, and MOR209/ES414, are currently being evaluated in clinical studies conducted on patients with mCRPC [226,279].

#### 5.1.13. STEAP1-Specific ICE

AMG 509 is a CD3- and STEAP1-specific bsAb-based T-cell engager that is used for the treatment of patients with STEAP1-expressing prostate cancers. A phase I clinical study (NCT04221542) on AMG 509 is currently being conducted to evaluate its safety, tolerability, PK, and efficacy in patients with mCRPC [280].

#### 5.1.14. SSTR2-Specific ICE

Tidutamab (XmAb18087) is a bsAb-based T-cell engager against CD3 and SSTR2 for the treatment of patients with advanced NETs [281]. A phase I/II clinical study (NCT04590781) is currently being conducted to evaluate the safety, tolerability, PK, and potential efficacy of tidutamab in patients with extensive-stage SCLC [256].

### 5.2. ICEs Targeting Tumor Antigens on Hematological Cancers

#### 5.2.1. BCMA-Specific ICE

Teclistamab (JNJ-64007957) is a CD3- and BCMA-specific bsAb-based T-cell engager based on the DuoBody platform [282]. In a phase I clinical study (NCT03145181), patients with R/R MM received teclistamab monotherapy at the RP2D. Of the 40 patients, 58% showed a very good PR (VGPR) or better outcome, and 30% showed a CR [283]. A phase III clinical study (NCT05083169) aimed at evaluating the efficacy of teclistamab in combination with daratumumab, which is a US FDA-approved anti-CD38 mAb, in the treatment of patients with R/R MM is ongoing [284]. Other CD31- and BCMA-specific bsAb-based T-cell engagers, such as elranatamab (PF-06863135), linvoseltamab (REGN5458), REGN5459, EMB-06, CC-93269 (EM801), TNB-383B, and AMG 701, are being actively assessed in various clinical studies [251,285].

#### 5.2.2. CD19-Specific ICE

Blinatumomab (Blincyto) is a CD3- and CD19-specific bsAb-based T-cell engager that was approved by the US FDA on the basis of the BiTE platform [286]. In a phase II/III clinical study (NCT02910063), patients with R/R aggressive B-cell NHL received blinatumomab as the second salvage therapy. Of the 41 patients, 9 showed complete metabolic responses, and 6 showed partial metabolic responses [287,288]. Then, in a phase II clinical study (NCT03023878), patients with high-risk DLBCL who received first-line treatment with rituximab plus chemotherapy were treated with blinatumomab. Of them, 7 of 8 patients with PR and SD after treatment with rituximab plus chemotherapy showed CR following treatment with one cycle of blinatumomab [289]. In addition, TNB-486—another anti-CD19 bsAb-based T-cell engager—is currently being evaluated in a phase I clinical study (NCT04594642) for its safety and clinical activity in patients with R/R B-cell NHL [290].

#### 5.2.3. CD20-Specific ICE

Mosunetuzumab (RG7828) is a CD3- and CD20-specific bsAb-based T-cell engager that is based on the CrossMab platform [291]. In a phase I/II clinical trial (NCT02500407), 90 patients with R/R FL who received ≥2 prior lines of therapy were treated with mosunetuzumab monotherapy. Of them, 57.8% showed a CR. The 12-month event-free rates were 80.1% in CR patients [292]. A phase III clinical study (NCT04712097) is currently being conducted to evaluate the safety and efficacy of mosunetuzumab in combination with lenalidomide in patients with R/R FL [269]. Additionally, other CD3- and CD20-specific bsAb-based T-cell engagers, such as glofitamab (RG6026 and RO7082859), epcoritamab (GEN3013), odronextamab (REGN1979), and plamotamab (XmAb13676), are being actively assessed in various clinical studies [293,294].

#### 5.2.4. CD22-Specific ICE

JNJ-75348780 is a CD3- and CD22-specific bsAb-based T-cell engager. It has been evaluated in a phase I clinical study (NCT04540796) to assess its safety and RP2D in patients with R/R B-cell NHL and CLL [295].

#### 5.2.5. CD30-Specific ICE

AFM13 is a bsAb-based NK-cell engager against CD16A and CD30 [296]. In a phase I/II clinical study (NCT04074746), patients with R/R HL and NHL were treated with precomplexed cord-blood NK cells with AFM13. Of the 13 patients, 8 showed a CR and 5 showed a PR after two cycles of treatment. Of 8 patients who showed a CR, 7 remained in CR at the median follow up of 6.5 months [297]. A phase II clinical study (NCT04101331) is currently being conducted to evaluate AFM13 monotherapy in patients with R/R peripheral T-cell lymphoma or transformed mycosis fungoides that are classified as a subtype of NHL [217].

#### 5.2.6. CD33-Specific ICE

AMG 330 is a CD3- and CD33-specific bsAb-based T-cell engager. A phase I clinical study (NCT02520427) enrolled 40 patients with R/R AML to receive AMG 330 monotherapy. Of them, two showed a CR, two showed a CR with incomplete hematologic recovery (CRi), and one achieved the morphologic leukemia-free stage (MLFS) [298]. In addition, JNJ-67571244—another CD3- and CD33-specific bsAb-based T-cell engager—has been included in a phase I clinical study (NCT03915379) to evaluate its safety and clinical activity in patients with R/R AML, as well as those with high-risk or very-high-risk myelodysplastic syndrome [299].

#### 5.2.7. CD38-Specific ICE

ISB 1342 (GBR 1342) is a CD3- and CD38-specific bsAb-based T-cell engager. A phase I clinical study (NCT03309111) is currently being conducted to evaluate the safety and efficacy of ISB 1342 in patients with R/R MM [300].

#### 5.2.8. CD123-Specific ICE

Flotetuzumab (MGD006) is a bsAb-based T-cell engager against CD3 and CD123 that is based on the DART platform [301]. In a phase I/II clinical study (NCT02152956), patients with primary induction failure (PIF) or early relapsed (ER) AML received flotetuzumab monotherapy. Among the patients with PIF/ER AML who were treated by using the RP2D, 31.8% showed a CR, CRi, and CR with partial hematologic recovery [302]. Flotetuzumab is in phase II clinical trial (NCT04582864) to evaluate its safety and efficacy in patients with relapsed AML following allogeneic hematopoietic-stem-cell transplantation [303]. Additionally, other CD3- and CD123-specific bsAb-based T-cell engagers, such as JNJ-63709178, vibecotamab (XmAb14045), and APVO436, are also being evaluated in clinical studies for their use in the treatment of CD123-expressing hematological malignancies [304].

#### 5.2.9. CLEC12A-Specific ICE

Tepoditamab (MCLA-117) is a bsAb-based T-cell engager against CD3 and CLEC12A [305]. A phase I clinical study (NCT03038230) enrolled 58 patients with AML to receive tepoditamab monotherapy; the result showed that >50% of bone-marrow blast reduction was observed in six patients, including one patient who achieved the MLFS [306].

#### 5.2.10. FLT3-Specific ICE

CLN-049 is a CD3- and FLT3-specific bsAb-based T-cell engager [307]. In preclinical studies, CLN-049 showed dose-dependent tumor reduction in the blood, and prolonged survival in a mouse xenograft model implanted with MOLM-13 human AML cells [308]. CLN-049 is currently being evaluated in a phase I clinical study (NCT05143996) for use in the treatment of patients with R/R AML [309]. In addition, AMG 427—another CD3- and FLT3-specific bsAb-based T-cell engager—is also in a phase I clinical study (NCT03541369) for use in the treatment of patients with R/R AML [310].

#### 5.2.11. GPRC5D-Specific ICE

Talquetamab (JNJ-64407564) is a bsAb-based T-cell engager against CD3 and GPRC5D that is based on the DuoBody platform. In a phase I clinical study (NCT03399799), the patients with R/R MM received talquetamab monotherapy at the RP2D, and 50% of the patients showed a VGPR or better outcome [311]. Talquetamab is currently being evaluated in a phase II clinical study (NCT04634552) for its safety and efficacy in patients with R/R MM [312].

## 6. Practical Considerations for the Development of Next-Generation ICEs

On the basis of the currently available evidence, this study proposes practical considerations for the development of the next-generation ICEs. First, the proximity of immune and tumor cells appears to underlie the improvement in the anticancer effects that are exerted by ICEs. A previous study compared (in vitro and in vivo) the efficacy of two different CD3 × GD2 ICEs that were designed by using the IgG–[L]–scFv2 or IgG–[H]–scFv2 format. The results show that IgG–[L]–scFv2 has an antitumor effect that is superior to that of IgG–[H]–scFv2 [221]. Second, another challenge in the ICE generation is to reinforce the PK of ICEs. Recently, several Fv-based ICEs, such as BiTE and DART, extend the in vivo half-life of antibodies. Thus, most bsAb-based ICEs are currently developed in the IgG-like or Fc-fused form [259,313]. Third, a high affinity of ICE for CD3 may not be necessary for achieving optimal T-cell activation without causing systemic toxicity. The systemic administration of anti-CD3 antibodies or catumaxomab (CD3 × EpCAM) often leads to systemic toxicity that is due to the uncontrolled T-cell activation in the peripheral blood [314]. Anti-CD3 antibodies of ICEs reportedly have various affinities that range from 1 to 200 nM [315]. Furthermore, a recent study reports that low-affinity anti-CD3 antibody shows better tumor distribution in vivo, without targeting T-cell-containing normal tissues, such as the spleen and lymph nodes [316]. Fourth, ICEs with silenced Fc are required for diminishing effector functions, such as ADCC, antibody-dependent T-cellular phagocytosis, and complement-dependent cytotoxicity. The binding of Fc to Fcγ receptors on immune cells may induce nonspecific immune activation that is associated with undesired toxicity [317]. Various ICEs with silenced Fc are currently being developed. For instance, the Fc domain of some ICEs was silenced by introducing various mutations: L234A-L235A-P329G (EM801), L234A-L235A-G237A-K322A (APVO436), E233P-L234V-L235A-G236del-S267K (Vibecotamab), N297G (AMG 330), or L234F-L235E-D265A (GEN3013) [11,247,318,319]. Furthermore, ICEs with silenced Fc improve the immune cells’ infiltration into solid tumors and enhance antitumor effects. Fifth, the optimal amino acid composition and ICE linkers’ length are the crucial factors for ensuring favorable physicochemical properties that are closely associated with manufacturing efficiencies [210]. Finally, despite recent advances in the identification of therapeutic targets of ICEs in cancer, cancers are highly heterogeneous diseases that harbor frequent mutations that are resistant to pre-existing drugs. Thus, the identification of novel potential targets for ICE therapy against cancers remains crucial for improving the clinical outcomes of patients with cancers.

## 7. Conclusions

Over several decades, cancer therapies have continuously evolved and have substantially improved the quality of life and the survival of patients with cancers. In particular, the advent of cancer immunotherapy has revolutionized the treatment paradigms in various cancers. Recently, bsAb-based ICEs have been established as a feasible modality for effective cancer immunotherapy and, consequently, have become the focus of several preclinical and clinical studies that are conducted globally to evaluate the efficacy and/or toxicity of bsAb-based ICEs to known and/or emerging therapeutic targets. There are still multiple challenges that are associated with the use of bsAb-based ICEs, such as drug resistance, manufacturing difficulties, and adverse effects on adjacent normal cells; nevertheless, these ICEs have potential in cancer treatments, as they are single-molecule anticancer agents that combine the advantages of two different mAbs that specifically target tumor-specific antigens or TAAs. Continuous research and development may help overcome the current pitfalls of these treatments, which may create new avenues for the successful treatment of patients with various cancers in the near future.

## Figures and Tables

**Figure 1 ijms-23-05686-f001:**
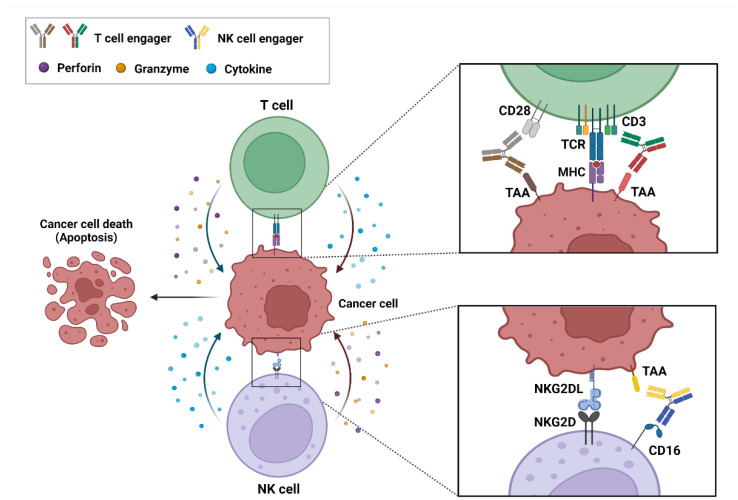
Action mechanism of bispecific antibody (bsAb)-based immune-cell engagers (ICEs) in cancers. The schematic drawing represents bsAb-based ICEs, including T- and natural killer-cell engagers, that bind simultaneously to tumor-associated antigens on cancer cells and specific antigens, such as CD3, CD28, and CD16 on immune cells. These interactions result in the formation of an immune synapse and the activation of immune cells that release cytokines, perforins, and granzymes to induce the cytotoxic effects on cancer cells.

**Figure 2 ijms-23-05686-f002:**
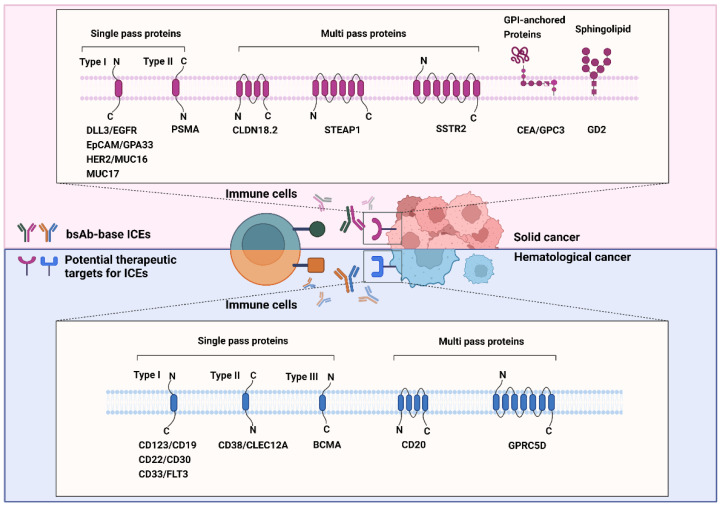
Known and emerging targets for immune-cell engager (ICE) therapy against cancers. The schematic representation shows known and emerging therapeutic targets of bispecific antibody-based ICEs in solid (red background) and hematological (blue background) cancers. All therapeutic targets listed in this figure are grouped on the basis of their relationship with the bilayer and transmembrane topology.

**Figure 3 ijms-23-05686-f003:**
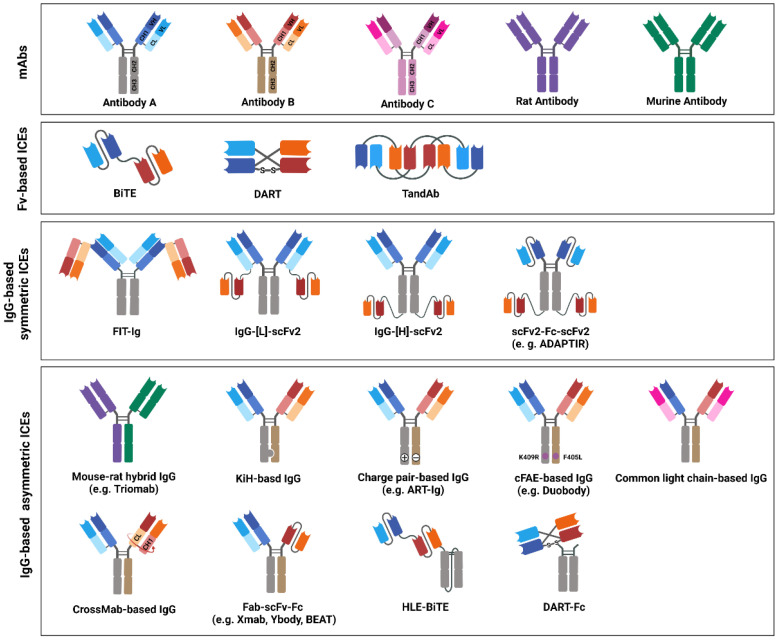
Immune-cell engager (ICE) structures in clinical studies, or ICEs approved by the United States Food and Drug Administration (US FDA) and/or European Medicines Agency (EMA)**.** The schematic drawing depicts the structures of bispecific antibody (bsAb)-based ICEs that are currently evaluated in clinical studies or that have been approved by the US FDA and/or EMA. The structures of bsAb-based ICEs are subdivided into three major classes: variable fragments -based, immunoglobulin G (IgG)-based symmetric, and IgG-based asymmetric ICEs. Variable heavy-chain domains (VHs) of two different antibodies, designated as antibodies A, B, or C, are shown in dark blue, red, or pink, respectively. The variable light-chain domains (VLs) are shown in light blue, red, and pink, respectively. Moreover, rat or mouse antibody is depicted in purple or green, respectively.

**Table 1 ijms-23-05686-t001:** Current statuses of bispecific antibody (bsAb)-based immune-cell engagers (ICEs) in clinical studies or bsAb-based ICEs approved by the United States Food and Drug Administration or European Medicines Agency.

BsAb Names	Company	Target	Role	Development Stage (Selected)
AFM13	Affimed	CD16 × CD30	NK cell engager	Phase II (NCT04101331)
AMG 160	Amgen	CD3 × PSMA	T-cell engager	Phase I (NCT03792841)
AMG 199	Amgen	CD3 × MUC17	T-cell engager	Phase I (NCT04117958)
AMG 330	Amgen	CD3 × CD33	T-cell engager	Phase I (NCT02520427)
AMG 427	Amgen	CD3 × FLT3	T-cell engager	Phase I (NCT03541369)
AMG 509	Amgen	CD3 × STEAP1	T-cell engager	Phase I (NCT04221542)
AMG 701	Amgen	CD3 × BCMA	T-cell engager	Phase I (NCT03287908)
AMG 910	Amgen	CD3 × CLDN18.2	T-cell engager	Phase I (NCT04260191)
APVO414/ES414/MOR209	Aptevo Therapeutics	CD3 × PSMA	T-cell engager	Phase I (NCT02262910)
APVO436	Aptevo Therapeutics	CD3 × CD123	T-cell engager	Phase I (NCT03647800)
Blinatumomab/Blincyto	Amgen	CD3 × CD19	T-cell engager	Marketed
Catumaxomab/Removab	Fresenius Biotechand and Trion Pharma	CD3 × EpCAM	T-cell engager	Withdrawn from the market
CC-1	University of Tubingen	CD3 × PSMA	T-cell engager	Phase I (NCT04104607)
CC-93269/EM801	Celgene	CD3 × BCMA	T-cell engager	Phase I (NCT03486067)
Cibisatamab/RG7802/RO6958688	Roche	CD3 × CEA	T-cell engager	Phase I (NCT02324257, NCT02650713)
CLN-049	Cullinan Oncology	CD3 × FLT3	T-cell engager	Phase I (NCT05143996)
Elranatamab/PF-06863135	Pfizer	CD3 × BCMA	T-cell engager	Phase II (NCT04649359)
EMB-06	EpimAb Biotherapeutics	CD3 × BCMA	T-cell engager	Phase I/II (NCT04735575)
Epcoritamab/GEN3013	Genmab A/S	CD3 × CD20	T-cell engager	Phase III (NCT04628494)
ERY974	Chugai	CD3 × GPC3	T-cell engager	Phase I (NCT05022927)
Flotetuzumab/MGD006	MacroGenics	CD3 × CD123	T-cell engager	Phase II (NCT04582864)
Glofitamab/RG6026/RO7082859	Roche	CD3 × CD20	T-cell engager	Phase III (NCT04408638)
ISB 1342/GBR 1342	Ichnos Sciences	CD3 × CD38	T-cell engager	Phase I (NCT03309111)
JNJ-63709178	Johnson & Johnson	CD3 × CD123	T-cell engager	Phase I (NCT02715011)
JNJ-63898081	Johnson & Johnson	CD3 × PSMA	T-cell engager	Phase I (NCT03926013)
JNJ-67571244	Johnson & Johnson	CD3 × CD33	T-cell engager	Phase I (NCT03915379)
JNJ-75348780	Johnson & Johnson	CD3 × CD22	T-cell engager	Phase I (NCT04540796)
Linvoseltamab/REGN 5458	Regeneron Pharmaceuticals	CD3 × BCMA	T-cell engager	Phase I/II (NCT03761108)
M701	YZYBio	CD3 × EpCAM	T-cell engager	Phase I (NCT04501744)
M802	YZYBio	CD3 × HER2	T-cell engager	Phase I (NCT04501770)
MGD007	Macrogenics	CD3 × GPA33	T-cell engager	Phase I/II (NCT03531632)
Mosunetuzumab/RG7828	Genentech	CD3 × CD20	T-cell engager	Phase III (NCT04712097)
Nivatrotamab/Hu3F8-BsAb	Y-mAbs	CD3 × GD2	T-cell engager	Phase I/II (NCT04750239)
Odronextamab/REGN1979	Regeneron Pharmaceuticals	CD3 × CD20	T-cell engager	Phase II (NCT03888105)
REGN4018	Regeneron Pharmaceuticals	CD3 × MUC16	T-cell engager	Phase I/II (NCT03564340)
REGN5459	Regeneron Pharmaceuticals	CD3 × BCMA	T-cell engager	Phase I/II (NCT04083534)
REGN5678	Regeneron Pharmaceuticals	CD28 × PSMA	T-cell engager	Phase I/II (NCT03972657)
REGN7075	Regeneron Pharmaceuticals	CD28 × EGFR	T-cell engager	Phase I/II (NCT04626635)
Talquetamab/JNJ-64407564	Johnson & Johnson	CD3 × GPRC5D	T-cell engager	Phase II (NCT04634552)
Tarlatamab/AMG 757	Amgen	CD3 × DLL-3	T-cell engager	Phase II (NCT05060016)
Teclistamab/JNJ-64007957	Johnson & Johnson	CD3 × BCMA	T-cell engager	Phase III (NCT05083169)
Tepoditamab/MCLA-117	Merus	CD3 × CLEC12A	T-cell engager	Phase I (NCT03038230)
TNB-383B	TeneoOne	CD3 × BCMA	T-cell engager	Phase I (NCT03933735)
TNB-486	TeneoTwo	CD3 × CD19	T-cell engager	Phase I (NCT04594642)
TNB-585	Amgen	CD3 × PSMA	T-cell engager	Phase I (NCT04740034)
XmAb13676/Plamotamab	Xencor	CD3 × CD20	T-cell engager	Phase I (NCT02924402)
XmAb14045/Vibecotamab	Xencor	CD3 × CD123	T-cell engager	Phase I (NCT02730312)
XmAb18087/Tidutamab	Xencor	CD3 × SSTR2	T-cell engager	Phase I/II (NCT04590781)

## Data Availability

The data collected in this study are available from the corresponding author upon reasonable request.

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
