# Peer review of "Bispecific Antibody-Based Immune-Cell Engagers and Their Emerging Therapeutic Targets in Cancer Immunotherapy"

_ijms, 2022, doi:10.3390/ijms23105686_

Round 1

Reviewer 1 Report

This manuscript reviewed the current understandings and the furture of bispecific antibodies for cancer therapies. This manscript cover a variety of cancer fields including hematological malignancies and epithelial tumors. There is few papers reviewing this topic and this manuscript would contribute to understanding more about this important issue. 

I think this review is valuable to be published, but there are several points to be revised. 

First, there are a few typographical errrors throughout the manuscript. I suggest a language editing by an native English speaker. 

Second, the problems to be solved for the improvement of bi-specific antibodies is only written in conclusion part of this manuscript. These problems should be written in more former part of this manuscript, or make new part except from "conclusion" part. The conclusion part should contain the summary of this manuscript. 

Reviewer 2 Report

It is a detailed review of the chemistry behind the bispecific antibodies including BiTE, ICE along with the various constructs. They also list the various antibodies with the antigenic target constructs in clinical studies.

However would help to elaborate on areas that is furthest in terms of clinical data especially in CD20 (with an impending FDA approval of Mosnetuzumab), CD19 (Blina approval) and CD30 (AFM13). 

Would be helpful to dilneate the production challenges and timelines for the various constructs as they get closer to approvals.

Reviewer 3 Report

The manuscript received for review, stating only based on its title and summary, should be extremely actual, interesting, and useful for a large group of researchers. It tries to cover a very broad area of ​​research and to signal the existence of specific themes related to the bispecific antibody. There is currently a large amount of original works on this subject. Even though the authors cite over 400, they are still far from exhausting the topic. Therefore, we should answer the question: who is this publication intended for? Because if it is for a person who is just initially dealing with the subject of immunotherapy with the use of antibodies, it is written in a too short, even signalling way, not explaining respectively the essence of this therapeutic procedure. If, on the other hand, it is addressed to high-class specialists in this field, it is presented too concisely, only pointing to the existence of individual problems and relevant literature on them.

The article is full of variety abbreviations and symbols used in it, and although they are systematically and correctly explained, their number is overwhelming and does not make it easy to read by people without proper, detailed knowledge. These are specific symbols, used for very narrow groups of substances and in this amount, they can be an obstacle in the ongoing tracking of the content of the article.

Such a mentioned concise approach to presenting the discussed problems are also chapters with a volume of 1-3 sentences (especially visible in part 3). Moreover, the proportion between the volume of the text of the article itself and the similar volume of the list of the quoted references is also striking.

Moreover, in a publication of this kind, it asks for the use of illustrative material in a greater extent, primarily in the form of various types of diagrams and schema.

Summarizing, I believe that this interested article would fulfil its function much better if it were divided into at least two independent parts, appropriately enriched in content. One of a more general nature and somewhat explanatory style - written in simple language. And the second, or even two seconds, more detailed ones, discussing the individual classes of the bispecific antibody in cancer immunotherapy, giving even more detailed information in the text than is currently the case. During these changes, the possible further expansion of the list of references should be limited.

In the conclusion from the above statement, it should be stated that the reviewed article is formally correct and could be accepted for publication in the presented form, but for its accessibility for the reader and maintaining the appropriate level of the Journal, it would be good to modify it according to the above-mentioned suggestions.
